# Coxsackievirus A2 Leads to Heart Injury in a Neonatal Mouse Model

**DOI:** 10.3390/v13081588

**Published:** 2021-08-11

**Authors:** Wangquan Ji, Peiyu Zhu, Ruonan Liang, Liang Zhang, Yu Zhang, Yuexia Wang, Weiguo Zhang, Ling Tao, Shuaiyin Chen, Haiyan Yang, Yuefei Jin, Guangcai Duan

**Affiliations:** 1Department of Epidemiology, College of Public Health, Zhengzhou University, Zhengzhou 450001, China; ayq116814@gs.zzu.edu.cn (W.J.); zpy6860@gs.zzu.edu.cn (P.Z.); liangruonanlucy@gs.zzu.edu.cn (R.L.); 202022272014960@gs.zzu.edu.cn (L.Z.); 202022272014965@gs.zzu.edu.cn (Y.Z.); 202012272014825@gs.zzu.edu.cn (Y.W.); sychen@zzu.edu.cn (S.C.); yhy@zzu.edu.cn (H.Y.); 2Department of Immunology, Duke University Medical Center, Durham, NC 27710, USA; wzhang033@icloud.com; 3School of Public Health, Xinxiang Medical University, Xinxiang 453003, China; 141050@xxmu.edu.cn; 4Henan Key Laboratory of Molecular Medicine, Zhengzhou University, Zhengzhou 450001, China

**Keywords:** hand, foot and mouth disease, coxsackievirus A2, matrix metalloproteinase, heart injury

## Abstract

Coxsackievirus A2 (CVA2) has emerged as an active pathogen that has been implicated in hand, foot, and mouth disease (HFMD) and herpangina outbreaks worldwide. It has been reported that severe cases with CVA2 infection develop into heart injury, which may be one of the causes of death. However, the mechanisms of CVA2-induced heart injury have not been well understood. In this study, we used a neonatal mouse model of CVA2 to investigate the possible mechanisms of heart injury. We detected CVA2 replication and apoptosis in heart tissues from infected mice. The activity of total aspartate transaminase (AST) and lactate dehydrogenase (LDH) was notably increased in heart tissues from infected mice. CVA2 infection also led to the disruption of cell-matrix interactions in heart tissues, including the increases of matrix metalloproteinase (MMP)3, MMP8, MMP9, connective tissue growth factor (CTGF) and tissue inhibitors of metalloproteinases (TIMP)4. Infiltrating leukocytes (CD45^+^ and CD11b^+^ cells) were observed in heart tissues of infected mice. Correspondingly, the expression levels of inflammatory cytokines in tissue lysates of hearts, including tumor necrosis factor alpha (TNF-α), interleukin-1beta (IL-1β), IL6 and monocyte chemoattractant protein-1 (MCP-1) were significantly elevated in CVA2 infected mice. Inflammatory signal pathways in heart tissues, including phosphatidylinositol 3-kinase (PI3K)-AKT, mitogen-activated protein kinases (MAPK) and nuclear factor kappa B (NF-κB), were also activated after infection. In summary, CVA2 infection leads to heart injury in a neonatal mouse model, which might be related to viral replication, increased expression levels of MMP-related enzymes and excessive inflammatory responses.

## 1. Introduction

Enterovirus (EV) 71 and several coxsackievirus (CV) serotypes (CVA2, A4, and A10) are frequently implicated in hand-foot-mouth disease (HFMD). Among the pathogens of HFMD, CVA2 has emerged as an active pathogen that has been implicated in HFMD and herpangina outbreaks worldwide such as China [1,2,3,4,5,6,7], Thailand [8], France [9], Vietnam [10] Brazil [11], and Korea [12,13], representing great socio-economic burden for the patient and society.

CVA2 was previously thought to cause sporadic infections with benign clinical presentation. A retrospective study compared the difference between CVA2 and EV71, which indicated that some CVA2-infected patients had central nervous system (CNS) complications and cardiopulmonary failure [14]. Moreover, severe complications or deaths associated with CVA2 have also been reported before [7,9,15,16]. A natural recombinant of CVA2 was found in 4 children with respiratory symptoms in Hong Kong, China, during the summer of 2012 and two of these children died [17]. Viral myocarditis is the most prevalent type of myocarditis with inflammatory response and an important contributor to dilated cardiomyopathy worldwide [18]. Severe myocarditis is one of the most important causes of death in CVA2 infections [9], which may be a sequelae of nervous system damage, although it is an uncommon serotype rarely associated with myocarditis. A majority of EV myocarditis survivors develop serious cardiac sequelae such as dilated cardiomyopathy [19], resulting from extracellular matrix remodeling after virus infection and immune-mediated injury. The matrix metalloproteinases (MMPs) have been implicated in cardiac repair and extracellular matrix (ECM) regulation after heart injury or stress [20,21]. Meanwhile, MMPs are important immune modulators affecting cytokine processing and cell migration through their matrix-degrading functions [22]. The alteration of endogenous tissue inhibitors of metalloproteinases (TIMPs) can directly inhibit MMP activity [23]. The interplay between MMPs and TIMPs has a regulatory role in myocardial remodeling and immune response [24]. In previous clinical studies [25] and autopsy [26], it was found that some patients with severe HFMD had serious complications, such as heart injury. Heart injury caused by enterovirus A serotype has been reported in previous animal models of HFMD, such as EV71 [27], CVA16 [28] and so on. The pathogenic mechanisms of different serotypes are still unclear. The clinical manifestations, epidemiology, and outcomes of the patients with CVA2 infection had not been reported in detail recently. These findings prompted us to study the heart injury after CVA2 infection.

In the present study, we applied a neonatal mouse model of CVA2 infection to investigate heart injury. We found CVA2 infection could cause heart injury through replication of the virus, increased expression levels of MMP-related enzymes and excessive inflammatory cytokines. Our study may provide a useful tool and hint for future research on the pathogenesis and therapeutic strategies for CVA2-associated heart injury.

## 2. Materials and Methods

### 2.1. Cells and Viruses

Human rhabdomoma (RD) cells were cultured in DMEM (Thermo Fisher Scientific Inc., Waltham, MA, USA) supplemented with 10% fetal bovine serum (Thermo Fisher Scientific Inc., Waltham, MA, USA) and incubated at 37 °C with 5% CO_2_. The CVA2 strain (HN202009, accession number: MT992622) was used for establishing animal model. The titers were determined through the 50% tissue culture infectious dose (TCID_50_) assay in accordance with the method of Reed and Muench [29]. All CVA2 stocks were subjected to three freeze-thaw cycles, clarified by centrifugation at 4000× *g* for 10 min at 4 °C, filtered through a 0.22 µm micron filter, and stored at −80 °C. The virus titer used in this study was 2.45 × 10^7^ TCID_50_/mL.

### 2.2. Mice and Infection Model

The BALB/c mice used in this study were obtained from Experimental Animal Center of Zhengzhou University, and all mice were housed in individually ventilated cages (IVC, Tecniplast) in a specific pathogen-free facility of the College of Public Health of Zhengzhou University on a 12 h light/dark cycle with ad libitum access to food and water. In the present study, we introduced a neonatal mouse model of CVA2 infection based on 5-day-old BALB/c mice inoculated with a lethal dose of CVA2 strain (10^4^ TCID_50_/mouse) via intramuscular (i.m.) route (*n* = 10 per group) [30]. In concurrence with the experimental groups, the control mice were sacrificed at 3 or/and 7 days post-infection (dpi), respectively. The heart tissues of all mice were either fixed in 4% paraformaldehyde and embedded in paraffin for histology and immunofluorescent staining (*n* = 4 per group) or snap-frozen in liquid nitrogen and stored at −80 °C for RNA isolation or protein extraction (*n* = 4 per group).

### 2.3. Immunofluorescent Staining

As mentioned above, after fixation, paraffin-embedded hearts were cut into 5 μm sections and the slices were stored at 4 °C until performing immunofluorescence (IF) staining. The expression of VP1 in cardiomyocytes was detected by immunofluorescence double-labeling staining. Immunofluorescence staining and scanning technical service were provided by Servicebio Biotech Co., Ltd. as described previously [31]. Fluorescence intensity was analyzed using Image J software.

### 2.4. Quantitative RT-PCR (qRT-PCR)

Heart tissues were homogenized, and total RNA was extracted using a TRIzol reagent (Thermo Fisher Scientific Inc., Waltham, MA, USA). Extracted RNA was reverse transcribed into cDNA using a relevant kit (Yeasen Biotechnology (Shanghai) Co., Ltd., Shanghai, China). The transcription level of VP1 and other genes (connective tissue growth factor, CTGF; B-type natriuretic peptide, BNP; lactate dehydrogenase, LDH; creatine kinase, CK; aspartate transaminase, AST; cardiac troponin I, CTNI) were analyzed by qRT-PCR using the instrument (Serial No. q225-0207, Kubo Tech Co., Ltd., Beijing, China). The primers used in this study were listed in Table 1. Relative mRNA levels were calculated using the 2^−ΔΔCt^ method.

### 2.5. ELISA

The concentrations of total proteins were determined using the BCA assay (Beijing Biomed Gene Technology Co., Ltd., Beijing, China) according to the manufacturer’s instructions. The levels of tumor necrosis factor alpha (TNF-α), interleukin (IL)-1β, IL-6 and monocyte chemoattractant protein-1 (MCP-1) in heart lysates were measured with corresponding ELISA kits (Biolegend, San Diego, CA, USA) and the results were normalized with the corresponding concentrations of total proteins.

### 2.6. Detection of Apoptosis

Apoptosis was detected in heart slices using the terminal transferase-mediated DNA nick end labeling (TUNEL) assay. Red staining indicated TUNEL-positive cells.

### 2.7. Tissue Histopathology

At 7 dpi, heart tissues were collected, fixed by formalin, embedded by paraffin sectioned, and stained with Masson’s trichrome.

### 2.8. Examination of Cardiac Enzymes

Levels of AST, CK, and LDH in the supernatants of heart homogenates were detected using relevant kits (Nanjing Jiancheng Bioengineering Institute, Nanjing, China) according to the manufacturer’s instructions.

### 2.9. Western Blot (WB)

Total proteins of mice hearts were extracted using a protein extraction kit (CWbio Company Ltd., Beijing, China) according to the manufacturer’s instructions. The protein samples were separated by 10% SDS-PAGE and electro-transferred onto PVDF membranes. The membranes were blocked with 5% defatted milk that was dissolved in PBST (phosphate-buffered saline, pH 7.6, containing 0.05% Tween20) for 1 h at room temperature. Then the membranes were incubated with primary antibodies (mAb) overnight at 4 °C, and then were washed thrice. After washing, the membranes were incubated with second antibodies for 1 h at room temperature. Next, the membranes were finally washed thrice and developed with an ECL enhanced Chemiluminescence Kit (Absin Bioscience, Inc., Shanghai, China).

### 2.10. Antibodies

The following primary antibodies were used in this study: anti-CTGF, TIMP-4, MMP9, CTNI (Proteintech Group, Inc., Wuhan, China); anti-MMP8, MMP3 (Abways Biotechnology Co., Ltd., Shanghai, China); anti-BNP (Cloud-Clone Technology Co., Ltd., Wuhan, China); anti-pp65, p65, pERK1/2 (Thr202/Tyr204), CD45, pIκBα, IκBα, pJNK, JNK2, ERK1/2, pp38, p38, AKT, pAKT (Thr308), anti-mouse IgG and anti-rabbit IgG (Cell Signaling Biotechnology, Inc., MA, USA); anti-CD11b (Abcam Biotechnology, Inc., Cambridge, UK); anti-CVA2 VP1 antibody (prepared in our own laboratory).

### 2.11. Statistical Analysis

Statistical analysis was performed with GraphPad Prism version 8.3 (San Diego, CA, USA). The results were expressed as the mean ± standard deviation (SD). The Mantel-Cox log rank test was used to compare the survival of mice in different groups. Unpaired student *t*-test or one-way analysis of variance (ANOVA) was employed to determine the significance of differences between groups. A *p*-value less than 0.05 was considered statistically significant in this study.

## 3. Results

### 3.1. CVA2 Infection Led to Heart Injury in a Mouse Model

Five-day-old BALB/c mice were inoculated with a lethal dose of CVA2 strain via intramuscular (i.m.) route. All infected mice died between 3 and 10 dpi. The control mice stayed healthy throughout the experiment (Figure 1A). To investigate viral replication, we measured CVA2 VP1 mRNA in heart tissues by qRT-PCR and detected CVA2 VP1 expression by IF. We found virus replication in heart tissues of infected mice at 3 and 7 dpi (Figure 1B). To confirm whether CVA2 could affect cardiomyocytes, we conducted immunofluorescence staining of viral VP1 proteins and cardiac troponin I (cardiomyocyte-specific antibody) and found VP1 signal was indeed located in cardiomyocytes (see white arrows) (Figure 1C). TUNEL assay revealed a mass of apoptotic cells in the hearts of infected mice (Figure 1D). Histopathological examination showed that inflammatory exudation (blue staining) among myocardial fibers began to increase gradually at 3 dpi and cardiac myocytes damage and interstitial edema were found at 7 dpi, which was a histological evidence of heart injury (Figure 1E). Together, our results indicated that CVA2 infection could cause heart injury in a neonatal mouse model.

### 3.2. Alterations of Cardiac Enzymes in CVA2-Induced Acute Heart Injury

As shown in Figure 2A, the transcription levels of BNP, AST, LDH, CTNI were significantly increased. Likewise, the activity of total AST and LDH in heart tissues of CVA2-infected mice was significantly higher than that in control mice (Figure 2B). Immunofluorescence staining revealed that the expression levels of BNP and CTNI in heart tissues were elevated, compared to control mice (Figure 2C,D). The protein expression level of CTNI in cardiomyocytes of CVA2-infected mice was also increased (Figure 2E). Together, our data indicated alterations of cardiac enzymes in CVA2-induced acute heart injury.

### 3.3. CVA2 Infection Led to the Disruption of Cell-Matrix Interactions in Heart Tissues

Disruption of cell-matrix interactions has devastating effects on the heart, and we next detected MMPs. The transcription levels of MMP3, MMP8, MMP9, CTGF and TIMP4 were significantly elevated in heart tissues of CVA2-infected mice (Figure 3A). We also used immunofluorescence staining (Figure 3B,C) and Western blot analysis (Figure 3D) to confirm the protein expression levels of above mediators. We found the expression levels of these proteins in hearts were increased by approximately two folds. Furthermore, Western blot analysis showed the higher expression of these proteins compared with control group, which was similar to the results of qRT-PCR and immunofluorescence staining (Figure 3D). Taken together, our results demonstrated that the heart injury induced by CVA2 was associated with the changes of extracellular matrix-related proteins.

### 3.4. Leukocyte Infiltration and Proinflammatory Cytokines Expression in Heart Tissues after CVA2 Infection

At 7 dpi, we determined leukocyte (CD45^+^) and monocytes (CD11b^+^) infiltration in heart slices of infected and control mice (Figure 4A). The numbers of CD45^+^ cells, as well as CD11b^+^, cells were significantly increased in infected mice, compared to control mice. Correspondingly, the expression levels of proinflammatory cytokines in tissue lysates, including IL-6, TNF-α, IL-1β, and MCP-1, were significantly elevated in CVA2 infected mice compared to control mice at 7 dpi (Figure 4B). We further analyzed inflammation-related signaling pathways in heart tissues by Western blot (Figure 4C). Phosphorylation (activation) of Akt, p65, JNK, IκBα, ERK1/2, and p38 were enhanced at 3 dpi and/or 7 dpi. Taken together, our results suggested that CVA2 infection led to leukocyte infiltration and inflammatory responses in heart tissues.

## 4. Discussion

Enteroviruses (EV) are ubiquitous human pathogens, which includes the subgroups enteroviruses, polioviruses, coxsackieviruses (A and B), and echoviruses [32]. CVA2 has emerged as an active pathogen that is attributed to HFMD and herpangina outbreaks worldwide. Generally, most CVA2 infections are mild and self-limiting [14], however some infections develop into severe complications such as myocarditis, CNS complications, pulmonary edema and even death [9,15]. EV-infected brainstem is generally thought to be responsible for cardiopulmonary collapse. However, we detected CVA2 antigen at multiple organs, including heart and observed heart injury in our previous mouse model of CVA2. These data suggested that heart injury could be one plausible cause of death in this mouse model. In the present study, we investigated possible mechanisms of CVA2-induced acute heart injury, which would be beneficial for understanding the pathogenesis of CVA2 infection and providing potential therapeutic targets for CVA2 infections.

Firstly, we repeated our previous animal model of CVA2, and all infected mice died within 10 days. The main cause of death may be complications such as pulmonary edema or encephalitis, but the contribution of viral heart injury should not be ignored. We detected CVA2 VP1 mRNA and antigen in heart tissues of infected mice, particularly at the later phase. Meanwhile, a mass of apoptotic cells in heart tissues of infected mice were observed by TUNEL assay. EV can persistently proliferate in the permissive tissues of the susceptible host and ultimately reach the myocardium or blood vessels through circulatory, lymphangitic spread, or both [33,34]. Coxsackieviruses could infect primarily cardiomyocytes, and due to extensive virus replication, a rapid cytolysis of these cells occurs [35]. Additionally, EV proteases (2A of CVB3) can cleave host proteins resulting in the disruption of cytoskeleton in myocytes [36]. Our histopathologic examination found CVA2 infection led to heart injury characterized by increased inflammatory exudation, cardiac myocytes damage and interstitial edema. Cardiac enzymes are always used to evaluate the degree of heart damage. Our data indicated that CVA2 infection had a significant effect on cardiac enzymes. In summary, CVA2 can lead to severe heart injury in a neonatal mouse model.

Biomarkers of cardiac injury such as BNP, AST, CK and CTNI are usually detected in patients with acute myocarditis [37,38]. Levels of cardiac biomarkers in heart lysates, including BNP, AST and CK were elevated in infected mice indicating myocardial damage. Plasma BNP might be a sensitive and reliable biomarker for the detection of cardiac involvement in children with severe EV71 infections [39]. Clinical and experimental data have suggested that the increased level of CTNI is more common than that of CK in acute myocarditis [40]. Serum CTNI has high specificity in the diagnosis of myocarditis and is used to indicate the prognosis of the disease [41]. In our study, we found that the protein level of CTNI in the heart tissues of infected mice was slightly increased, but the transcription level of CTNI was significantly increased. The elevated activity of total AST and LDH was strong evidence of impaired cardiac protein regulation. It is worth noting that cardiac protein regulation may remain impaired despite reversible cardiac remodeling after acute viral myocarditis [42].

The damage and healing of the heart muscle is often accompanied by a change of the normal interstitial matrix, eventually resulting in chronic cardiac failure [43]. We observed increased transcriptional levels of MMP-3, MMP-8 and MMP-9, as well as CTGF, TIMP4 during heart injury induced by CVA2. The above results were verified by Immunofluorescent staining and Western blot analysis. As we all know, coxsackievirus B3 (CVB3) is the most prevalent pathogen that can cause viral myocarditis. Alterations of MMPs and TIMPs after CVB3 infection have been reported in adolescent mouse models [44]. Although the animal models of myocarditis are very different between CVB3 and CVA2, previous studies on CVB3 could still provide a reference for exploring pathogenic mechanisms of CVA2-related heart injury. MMPs expression in viral myocarditis has been linked to the onset of myocarditis and the long-term sequelae. MMPs are important regulators of the extracellular matrix, degrading all the different components of the matrix [43]. The transcription of many MMPs, especially MMP-3, MMP-8 and MMP-9 are upregulated in the acute phase of CVB3-induced myocarditis [45,46]. CTGF is a crucial molecule in the development of fibrosis in ongoing enteroviral myocarditis [47,48]. When considering MMPs activity during pathological states, we must take the TIMPs into account because an imbalance of TIMP level will affect the activity of MMPs and vice versa. All four TIMPs can inhibit the action of most of the 23 different MMPs [24]. Consequently, the TIMPs are considered as the important regulators during tissue remodeling. Many studies have reported the expression level of TIMP-4 is down-regulated in CVB3-induced myocarditis [45,49], which is in contrast with our result. Animal studies have suggested a role for TIMP4 in several inflammatory diseases and cardiovascular pathologies [50,51]. TIMP4 is most clearly visible in cardiovascular tissue areas populated by abundant inflammatory cells [52]. This viral myocarditis can be demarcated into three stages, which include sequentially viremia and direct virus injury, the inflammatory response, and variable degrees of reclamation of the myocardium with chamber dilatation [53]. Therefore, we speculated the high expression of TIMP4 was related to the inflammatory response at acute phase, which was confirmed by our findings that CVA2 infection led to abundant leukocyte infiltration in heart tissues. MMPs not only function in degradation of matrix but also act as major regulators of cytokine production, angiogenesis, cell migration and contributing to myocardial injury and remodeling during the progression of myocarditis [43,54]. For example, MMP8 and MMP9 regulate both innate and adaptive immunity through both proteolytic activation IL-1β and inactivation IFN-β [20,55,56]. MMP-9 has been reported to be secreted by immune cells aiding matrix degradation during migration [54]. The different proteases actions in the setting of viral myocarditis could tip the balance either to a beneficial immune response for the host, or to an exaggerated inflammatory reaction with detrimental results for tissue function and structure. Therefore, inhibition of MMP activity may be a potential approach against myocardial injury [57]. According to our findings, the changes of MMPs and TIMPs may be more related to the acute inflammatory response compared with CVB3 infection, and the role in matrix remodeling seems to be not particularly obvious, which needs further study and long-term observations.

The balance between the protective and deleterious effects of the immune and inflammatory responses was involved in the severity of viral myocarditis. At the acute phase, virus replication in the cardiomyocytes not only damages the heart directly but also induces a series of immunological responses [58]. Leukocyte infiltration in the heart is a critical reflection of host defense in response to viral infection, but an inappropriate and exaggerated immune response may also result in cardiac injury. Most tissue damage in myocarditis may be due to the interaction between the viral and the immune system. We observed markedly monocytes infiltration in heart tissues of infected mice. During the inflammatory phase, prominent cellular infiltration, including monocytes, enter the heart to clear the virus but may result in additional dissolution of the heart tissue [35]. Transfer of mononuclear cells from CVB3-infected mice or patients with myocarditis into genetically identical or immunodeficient mice, respectively, also exacerbates myocardial damage [59]. Correspondingly, the concentrations of inflammatory cytokines, including IL-6, TNF-α, IL-1β, and MCP-1, in heart lysates were significantly increased in CVA2 infected mice, which could result in heart injury [60]. CVB3 infection stimulates the expression of MCP-1 in myocardial cells, which subsequently leads to migration of mononuclear cells in viral myocarditis [61]. Emerging evidence has revealed that both t helper (Th) 1 cells are involved in the pathogenesis of viral myocarditis. Th1 cells and their proinflammatory cytokines, such as IL-1β and TNF-α, were shown to be etiological factors in the induction of viral myocarditis [62]. In addition to their role in modulating lymphocyte function, these proinflammatory cytokines also disturb the balance between MMPs and TIMPs, which are likely the major regulators of MMP activity [24]. Proinflammatory cytokines such as TNF-α and IL-1β secreted by immune cells may trigger further MMP up-regulation and aggravate heart injury.

We also explored inflammatory signal pathways that might be activated in the infected hearts. Western blot analysis revealed activation of PI3K-AKT [63], MAPK [64] and NF-κB signaling [65,66] in infected mice. The activation of these pathways is associated with the regulation of many inflammatory cytokines, including IL-6, TNF-α, IL-1β, and MCP-1. In EV-associated diseases, the activation of above signaling pathways can cause high expression of inflammatory cytokines [67], which is consistent with our results.

## 5. Conclusions

CVA2 infection leads to heart injury in a neonatal mouse model, which might be related to virus replication, alterations of MMP-related enzymes and excessive inflammatory responses. Our mouse model replicates many features of illness in acute heart injury, which may provide a useful tool for future research on CVA2pathogenesis and treatment strategies for heart injury induced by CVA2.

## Figures and Tables

**Figure 1 viruses-13-01588-f001:**
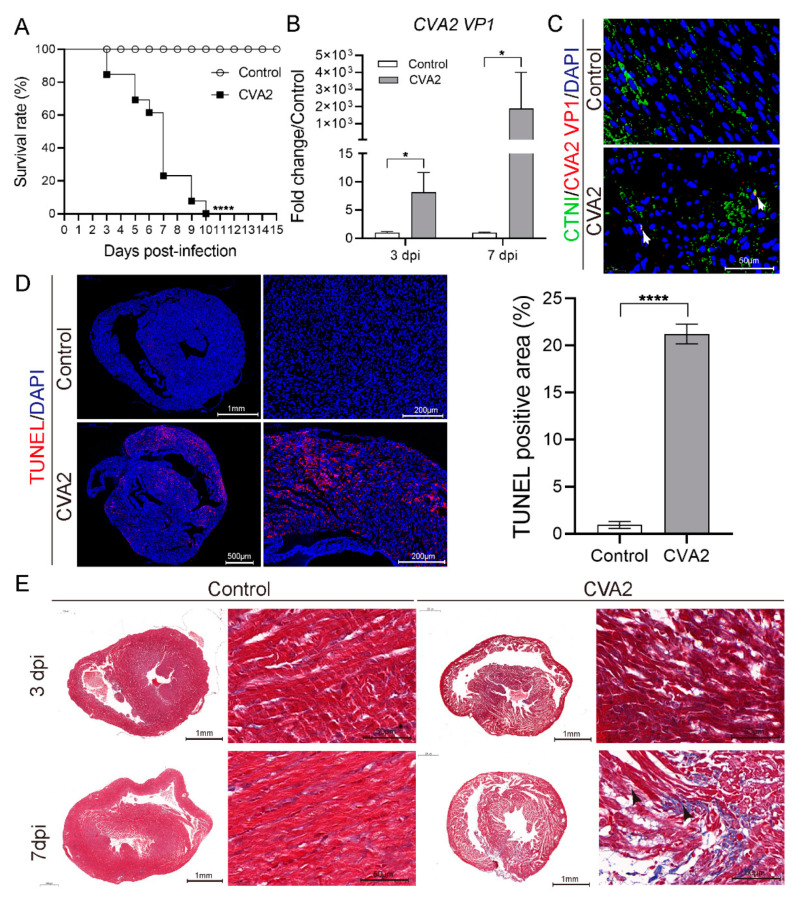
CVA2 infection leads to heart injury in a neonatal mouse model. (**A**) The survival rates of mice in two groups. (**B**) The transcription level of VP1 mRNA at 3 dpi and 7 dpi (*n* = 4 per group). (**C**) Colocation of VP1 protein (red) and CTNI (green) in heart slices and white arrows indicate infected cardiomyocytes (*n* = 4 per group). (**D**) Cardiomyocyte apoptosis was evaluated by TUNEL assay in heart slices (*n* = 4 per group). (**E**) Masson staining of the heart tissues was performed to evaluate the histological changes at 3 dpi and 7 dpi (*n* = 4 per group). The black arrows indicate the damage of cardiomyocytes. * *p* < 0.05, **** *p* < 0.0001 vs. control.

**Figure 2 viruses-13-01588-f002:**
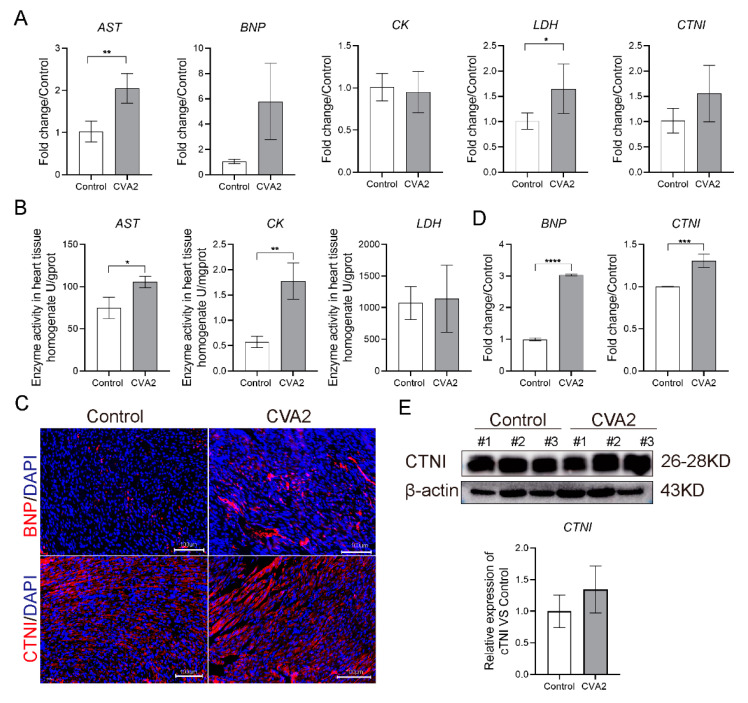
CVA2 infection increases the level of cardiac enzymes. (**A**) Quantification of mRNA levels of AST, BNP, CK, LDH and CTNI in the heart tissues at 7 dpi (*n* = 4 per group). (**B**) The activity of AST, CK and LDH in heart lysates at 7 dpi (*n* = 6 per group). (**C**) Immunofluorescence staining of BNP and CTNI (red) in the heart slices (*n* = 4 per group). (**D**) The fluorescence intensity of BNP and CTNI expression. (**E**) WB analysis of CTNI expression in heart tissues (*n* = 3 per group). Each bar indicates mean ± SD. * *p* < 0.05, ** *p* < 0. 01, *** *p* < 0.001, **** *p* < 0.0001 vs. control.

**Figure 3 viruses-13-01588-f003:**
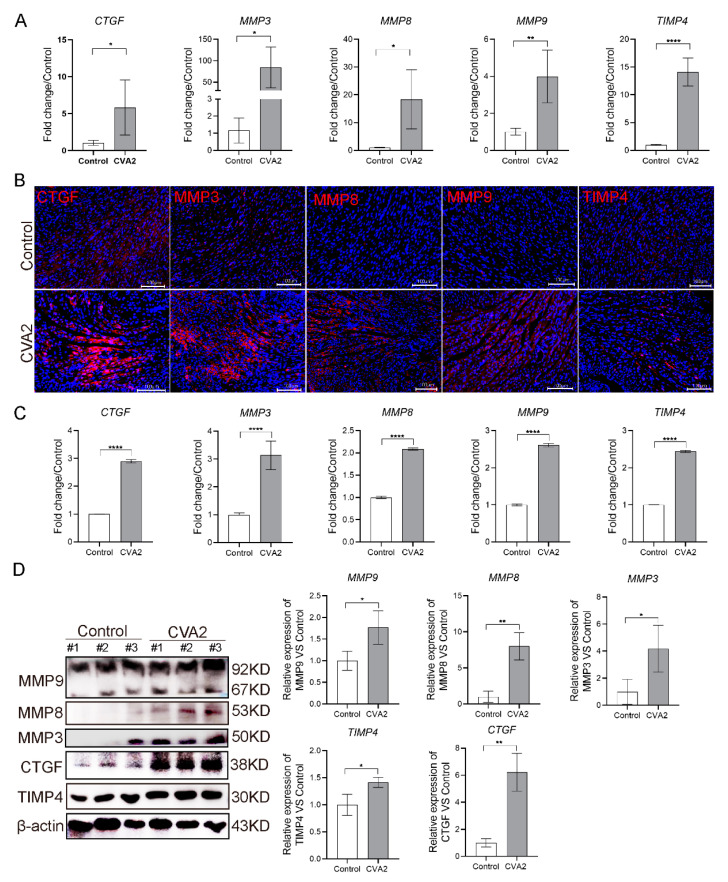
The disruption of cell-matrix interactions in heart tissues after CVA2 infection. (**A**) Relative expressions of CTGF, MMP3, MMP8, MMP9 and TIMP4 mRNA in heart tissues. (**B**,**D**) Immunofluorescence staining and WB analysis of CTGF, MMP3, MMP8, MMP9 and TIMP4 protein (red) in heart tissues. (**C**) The fluorescence intensity of CTGF, MMP3, MMP8, MMP9 and TIMP4 in mouse hearts (*n* = 4 per group). * *p* < 0.05, ** *p* < 0. 01, **** *p* < 0.0001 vs. control.

**Figure 4 viruses-13-01588-f004:**
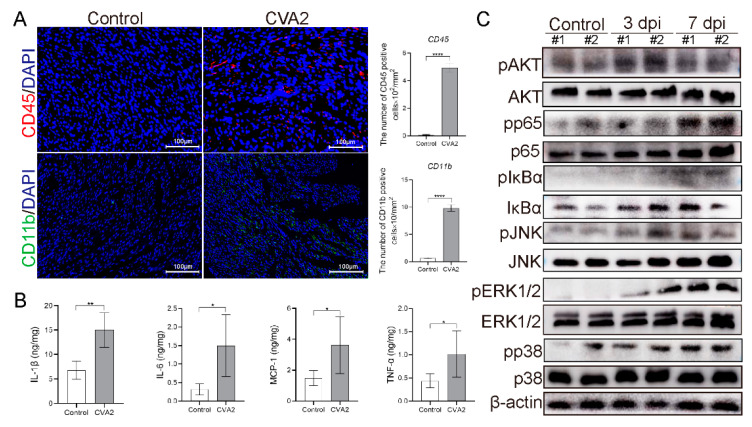
CVA2 infection induces excessive inflammatory responses in heart tissues. (**A**) Leukocyte infiltration in heart slices of control and infected mice. The numbers of CD45 or CD11b positive (CD45^+^ or CD11b^+^) cells (per mm^2^ heart tissues) were quantified by Image J soft-ware (*n* = 4 per group). (**B**) The concentrations of cytokines (IL-1β, IL-6, MCP-1 and TNF-α) in the heart tissues of mice were determined using ELISA assay (*n* = 4 per group). (**C**) Activation of inflammatory signal pathways (PI3K-AKT, MAPK and NF-κB) was evaluated by Western blotting. * *p* < 0.05, ** *p* < 0. 01, **** *p* < 0.0001 vs. control.

**Table 1 viruses-13-01588-t001:** Primers used in this study.

Gene	Forward (5’->3’)	Reverse (5’->3’)	Product Lengths (bp)
CVA2-VP1	TCAGTCCCATTCATGTCGCC	AATGCGTTGTTGGGGCATTG	118
CTGF	GGGAGAACTGTGTACGGAGC	AGTGCACACTCCGATCTTGC	97
MMP3	TCTCAGGTTCCAGAGAGTTAGA	TGTCACTGGTACCAACCTATTC	239
MMP8	TTGAGAAAGCTTTTCACGTCTG	CTTGAGACGAAAGCAATGTTGA	97
MMP9	CAAAGACCTGAAAACCTCCAAC	GACTGCTTCTCTCCCATCATC	105
TIMP4	GGCCGGAACTACCTTCTCACT	CACCCTCAGCAGCACATCTG	75
BNP	TGCTGGAGCTGATAAGAGAAAA	GAAGGACTCTTTTTGGGTGTTC	96
LDH	GGGAAAGTCTCTGGCTGATGAA	CTGTCACAGATATACTTTATCGGC	140
CK	AAACCCACAGACAAGCATAAGA	CTCTTCAGAGGGTAGTACTTGC	233
AST	ACAAGAACACACCAATCTACGT	ATAGGGCCGAATGTCCTTAAAA	92
CTNI	CACACGCCAAGAAAAAGTCTAA	GCATAAGTCCTGAAGCTCTTCA	194
Mouse β-actin	GTGCTATGTTGCTCTAGACTTCG	ATGCCACAGGATTCCATACC	174

## Data Availability

All data generated or analyzed during this study are included in this published article.

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
