# Peer review of "Coxsackievirus A2 Leads to Heart Injury in a Neonatal Mouse Model"

_viruses, 2021, doi:10.3390/v13081588_

Round 1

Reviewer 1 Report

This paper is an extension of a previous publication by Drs. Jin and Duan and colleagues (Ji, W, et al. 2021 Front. Microbiol. 12:658093) establishing a neonatal murine model of coxsackievirus A2 (CVA2)  The previous work established the ability of an RD cell-adapted viral strain to induce of disease in the neonatal BALB-c mouse, and demonstrated dosage, method of inoculation and survival. In the previous study they demonstrated a high amount of skeletal muscle myolysis and high viral load in this tissue but also pathology and viral titer in several tissues including the heart. In this previous work they also established a high level of  IL-6, IL-10 and MCP-1 in brain, lung, and muscle.  This paper extends this model in examining the heart in greater detail.  I am surprised that the authors did not cite the previous study in the background of the paper as questions that arise in such a model study are the choice of dosage, route of inoculation and mortality and were addressed in the previous publication. Was the intramuscular inoculation route chosen to maximize cardiac infection and pathology? This should be discussed as well.

The use of the neonatal mouse in generating human enterovirus A disease models has been examined for several the serotypes of this species, such as enterovirus A71, coxsackieviruses A4, A6, A10 and A16.  As the neonatal mouse allows extensive infection, it is understandable that this must be the first murine model used for viruses in which the receptor is unknown. However, the authors need to address the differences that a neonatal model generates.  Neonatal models have several issues in examining enteroviral diseases outside severe systemic neonatal enterovirus infections due to the high-level systemic infection and consequent high level innate immune response and multiple organ pathology.  This needs to be addressed in the discussion as in several places the authors compare their results with other enteroviral murine models in which older mice were infected.  For example, all neonatal mouse models (particularly a lethal model) have a high induction of IL-6 and other inflammatory cytokines and to some extent these should be present in the heart as long as the infection is ongoing.  And I note that this was true for other tissues in the previous study by authors looking at lung, muscle, and brain. This was also true for lymphocyte infiltration in other tissues in the previous study.

This is not to say that the work is without value, the authors need to emphasize the fact that they are examining the effects of infection in the heart by an enterovirus A serotype as little work has been done looking at the cardiac effects in a murine model of these serotypes. How does this model of CVA2 compare with similar neonatal studies of enterovirus 71? If enterovirus 71 cardiac effects have not been addressed in these studies, the authors should discuss this. 

The demonstration of CVA2 in the heart at a time in which apoptosis is also present as well as the effects of this severe infection on matrix metalloproteinases and tissue inhibitors of metalloproteinases is of interest but again, should be discussed because this is a severe infection of neonates unlike most of the coxsackievirus B3 studies in which adolescent mice are infected.  In lines 318-319, one important difference in the CVB3 study was the use of adolescent mice infected at a non-lethal dosage.

In the Methods section, lines 89-99, please state the number of mice used in the infected and control groups for day 3 and day 7 P.I.  It is particularly important when you are not showing data points.

In Section 3.1, lines 163-172 and Figure 1B, I can see the fold change over the beta actin level in qRT-PCR data, but what does that correspond to in terms of viral copy number per cell or per gram of cardiac tissue?  This is necessary to provide comparison to other studies of enteroviruses in the heart. 

In Figure 3, the quality of the western blots (D) for MMP8, 9 and 3 and CTGF is so poor that they do not seem to correspond to the IF (B) and the fluorescence intensity data (C).  I note that the authors have given readings for each lane, but they really do not seem to be correct.  Could the authors provide a less intense version of the western blot, as I think background is making Figure 3D appear incorrect?

Minor points:

The authors have typos using the abbreviation EV71 instead of CVA2 on lines 192, 195 and 196.

There are several small errors of English usage.

This extension of a neonatal mouse model of CVA2 infection to explore cardiac effects provides interesting and useful information on how an enterovirus A serotype can cause cardiac disease.  However, since it is a neonatal model, the effects upon the heart of a lethal neonatal infection must be discussed when comparing this study to those of enterovirus B models using adolescent mice.  This will require rewriting of parts of the discussion.  There are three issues with the data provided that can be addressed. 

Author Response

Q1: This paper is an extension of a previous publication by Drs. Jin and Duan and colleagues (Ji, W, et al. 2021 Front. Microbiol. 12:658093) establishing a neonatal murine model of coxsackievirus A2 (CVA2). The previous work established the ability of an RD cell-adapted viral strain to induce of disease in the neonatal BALB-c mouse, and demonstrated dosage, method of inoculation and survival. In the previous study they demonstrated a high amount of skeletal muscle myolysis and high viral load in this tissue but also pathology and viral titer in several tissues including the heart. In this previous work they also established a high level of IL-6, IL-10 and MCP-1 in brain, lung, and muscle. This paper extends this model in examining the heart in greater detail. I am surprised that the authors did not cite the previous study in the background of the paper as questions that arise in such a model study are the choice of dosage, route of inoculation and mortality and were addressed in the previous publication. Was the intramuscular inoculation route chosen to maximize cardiac infection and pathology? This should be discussed as well.

When we wrote the current manuscript, the previous article had not been accepted. Thus, we did not cite the previous study in the background. Skeletal muscles are known to support persistent enterovirus infection and provide a viral source of entry into the CNS during poliovirus infection. In the previous study, we found after infecting via intramuscular injection the mice developed significant clinical signs, including weight loss, reduced mobility, ataxia, and limb paralysis, which were similar with the clinical signs of human infections. We have discussed the reason why we chose the intramuscular inoculation route rather than other routes in the previous study.

Q2: The use of the neonatal mouse in generating human enterovirus A disease models has been examined for several the serotypes of this species, such as enterovirus A71, coxsackieviruses A4, A6, A10 and A16.  As the neonatal mouse allows extensive infection, it is understandable that this must be the first murine model used for viruses in which the receptor is unknown. However, the authors need to address the differences that a neonatal model generates. Neonatal models have several issues in examining enteroviral diseases outside severe systemic neonatal enterovirus infections due to the high-level systemic infection and consequent high level innate immune response and multiple organ pathology.  This needs to be addressed in the discussion as in several places the authors compare their results with other enteroviral murine models in which older mice were infected.  For example, all neonatal mouse models (particularly a lethal model) have a high induction of IL-6 and other inflammatory cytokines and to some extent these should be present in the heart as long as the infection is ongoing.  And I note that this was true for other tissues in the previous study by authors looking at lung, muscle, and brain. This was also true for lymphocyte infiltration in other tissues in the previous study.

This is not to say that the work is without value, the authors need to emphasize the fact that they are examining the effects of infection in the heart by an enterovirus A serotype as little work has been done looking at the cardiac effects in a murine model of these serotypes. How does this model of CVA2 compare with similar neonatal studies of enterovirus 71? If enterovirus 71 cardiac effects have not been addressed in these studies, the authors should discuss this.

We agree with the reviewer, and have made extensive revisions to the manuscript. In previous clinical studies (PMID: 20543760) and autopsy (PMID: 22827767), it was found that some patients with severe hand, foot and mouth disease had serious complications of heart injury. The establishment of animal models needs to consider many factors, such as species, virus strain, dose, route, age, etc., but the ultimate purpose of establishing animal models is to better simulate the response of humans after natural infection of the virus as a substitute for experiments. The pathogenic mechanisms of different serotype enteroviruses are not completely the same. Heart injury caused by enterovirus A serotype has been reported in other animal models of hand, foot and mouth disease, such as EV71 (PMID: 31366973), CVA16 (PMID: 31351798) and so on. We have discussed relevant findings in other enterovirus A serotypes.

Q3: The demonstration of CVA2 in the heart at a time in which apoptosis is also present as well as the effects of this severe infection on matrix metalloproteinases and tissue inhibitors of metalloproteinases is of interest but again, should be discussed because this is a severe infection of neonates unlike most of the coxsackievirus B3 studies in which adolescent mice are infected. In lines 318-319, one important difference in the CVB3 study was the use of adolescent mice infected at a non-lethal dosage.

Those enteroviruses that tend to infect internal organs comprise three polioviruses, 23 coxsackieviruses group A, and 6 coxsackieviruses group B (CVB). Among the six serotypes of CVB, only three, CVB1, 3 and 5, are notably cardiotropic. As we all know, myocarditis is most commonly caused by viral infections of the heart, with coxsackievirus B3 (CVB3) being among the most prevalent pathogens. CVB3-related damage to the heart is induced both directly by virally mediated cell destruction and indirectly due to the immune and autoimmune processes reacting to virus infection (PMID: 25865198). As you mentioned, one important difference in the CVB3 study was the use of adolescent mice infected at a non-lethal dosage. In the present study, we applied the neonatal mice to establish the model. Although the animal models of myocarditis are very different between the two viral strains, previous studies on CVB3 can still provide a reference for studying pathogenic mechanism of CVA2-related heart injury.

Q4: In the Methods section, lines 89-99, please state the number of mice used in the infected and control groups for day 3 and day 7 P.I.  It is particularly important when you are not showing data points.

We have revised it. We stated the number of mice used in our experiments in figure legends.

Q5: In Section 3.1, lines 163-172 and Figure 1B, I can see the fold change over the beta actin level in qRT-PCR data, but what does that correspond to in terms of viral copy number per cell or per gram of cardiac tissue?  This is necessary to provide comparison to other studies of enteroviruses in the heart.

The fold change of CVA2 VP1 mRNA normalized by the b-actin was used to quantify the viral titers of heart tissues. This method was also used in other similar publications (PMID: 27499235; PMID: 32149091; PMID: 31366973).

Q6: In Figure 3, the quality of the western blots (D) for MMP8, 9 and 3 and CTGF is so poor that they do not seem to correspond to the IF (B) and the fluorescence intensity data (C). I note that the authors have given readings for each lane, but they really do not seem to be correct.  Could the authors provide a less intense version of the western blot, as I think background is making Figure 3D appear incorrect?

We have repeated the experiments or reduced the background, and provided more clear bands for MMP8, 9 and 3 and CTGF.

Minor points:

Q7: The authors have typos using the abbreviation EV71 instead of CVA2 on lines 192, 195 and 196.

We have revised it.

Q8: There are several small errors of English usage.

We have revised it.

Q9: This extension of a neonatal mouse model of CVA2 infection to explore cardiac effects provides interesting and useful information on how an enterovirus A serotype can cause cardiac disease.  However, since it is a neonatal model, the effects upon the heart of a lethal neonatal infection must be discussed when comparing this study to those of enterovirus B models using adolescent mice.  This will require rewriting of parts of the discussion. There are three issues with the data provided that can be addressed.

The reviewer certainly raised a valid point. We agree with the reviewer that the different animal models used in CVA2 and enterovirus B should be discussed in Discussion section. We have highlighted this difference and re-edited it in discussion.

Reviewer 2 Report

The proposed manuscript described how CVA2 leads to heart injury in a neonatal mouse model . This article is not original and does not add significant original data about the EV induced heart injury with classical markers, cytokines or MMPs investigation (Circulation AHA journal see 1995-2021 references). However the part concerning the EV induced vascular cell dysfunctions is more interesting and could be of major interest in a newly formatted article focusing on viral activation ans pathways of inflammation. A new manuscript with an extensive editing is recommended . Note that no ethical statements concerning mice model are given by the authors (authorization agreement references).

Author Response

Q1: The proposed manuscript described how CVA2 leads to heart injury in a neonatal mouse model. This article is not original and does not add significant original data about the EV induced heart injury with classical markers, cytokines or MMPs investigation (Circulation AHA journal see 1995-2021 references).

Our research is fundamentally different from others. CVA2 recently emerged as an important pathogen of hand, foot, and mouth disease (HFMD). Clinical findings suggest that most of HFMD patients are general mild, but some of patients can develop into more serious diseases, such as myocarditis. In this study, we studied the possible mechanisms of heart injury in a neonatal mouse model, which will be useful for understanding the pathogenesis of CVA2 infection. Although other studies have also studied heart injury and changes of cytokines and MMPs after viral infection, this does not mean that all viral infections share same pathogenesis. Compared with CVB3-induced fulminant myocarditis, the heart injury caused by CVA2 is mainly manifested in inflammation, rather than extracellular matrix changes.

Q2: However, the part concerning the EV induced vascular cell dysfunctions is more interesting and could be of major interest in a newly formatted article focusing on viral activation ans pathways of inflammation. A new manuscript with an extensive editing is recommended.

Thank you for your good suggestion. The results of vascular cell dysfunctions are related to extracellular matrix changes, such as MMPs activation. Inflammatory cytokine release and leukocyte infiltration are also linked to vascular cell dysfunctions. We believe that these results together provided a more comprehensive pathogenic mechanism.

Q3: Note that no ethical statements concerning mice model are given by the authors (authorization agreement references).

In the section of “Institutional Review Board Statement”, we have provided ethical statements concerning mice model (protocol code: ZZUIRB2020-29, date of approval: April 2019). 

Reviewer 3 Report

In this study Ji et al describe cardiac injury induced by CVA2 in a neonatal mouse model. The study is well written and contains several investigations confirming CVA2 induced damage of the heart and induction of myocarditis.

Points of critic

  1. There was a previous paper of the group published in Frontiers in Microbiology where authors used the same CVA2, variant as in the present manuscript. In this previous paper I tried to open the link for the virus HN202009, but found “ AL_NL060I15.R AL_NL Arabidopsis lyrata genomic 3', genomic survey sequence”. Moreover, I did not found an entry for the accession number MT992622 for HN202009. Please insert into the manuscript a link where the accession number can be found.
  2. Figure 1C I am not sure, whether the cells which show brown staining are cardiomyocytes. At least the left cell (left arrow) could also be a macrophage. For the right arrow I am not sure what it shows. Coxsackieviruses can also infect immune cells and other cells , e.g. endothelial cells. To confirm that the VP1 signal is indeed located in cardiomyocytes a co-staining with cardiomyocyte specific antibody should be done. Moreover, it would be helpful to examine whether cardiomyocytes are targets of the virus by infection of the cells in vitro with the CVA2 strain.
  3. Line 108 Please specify the instrument using for real-time PCR
  4. Line 192 to 196, 210. I believe the authors mean CVA2 and not EV71?
  5. Figure 2E the WB should be quantified and the data shown as diagram.
  6. Line 213 replace “cardiomyocytes” by heart, as the whole heart was investigated.
  7. Line 196 the authors state that the data indicate cardiac dysfunction. This term is typically used to describe impaired contractility of the heart, which was not determined here. Thus, another term should be use e.g. dysregulation of cardiac proteins. Please also replace “cardiac function” in line 189 by another therm.
  8. The histological data described in the text (line 172-176) should be better explained in the figure 1E . Please explain blue staining (day 7 CVA2). Is it collagen, does it show fibrosis? The authors mention occurrence of damage of cardiomyocytes. Insert arrows for such events.
  9. Although pathological alteration as shown in Fig. 1E seems to be not as prominent as at day 7 the data from day 3 should be described in the text.
  10. Line 220, replace Figure by Figure 3
  11. Figure 3D the WB seems not confirm some of the data obtained with qRT-PCR and IF, such as I do not see increase of MMP8 in CVA2-infected mice in WB, it seems rather to be decreased. TIMP4 expression levels are similar in untreated and CVA2-infected mice in WB, whereas it is distinctly higher expressed in CVA2-infected mice as shown by IF and qRT-PCR. As also the loading control is highly variable, a quantification of WB should be done and the data shown in a diagram.
  12. Line 229-234 Based on measurement of certain biomarkers of endothelial dysfunction the authors state that CVA2 infected mice have endothelial dysfunction. There are no clinical data determining endothelial dysfunctions. Accordingly, their statement should be adapted.
  13. Line 249. Authors state that inflammatory -related pathways were induced. From the WB for AKT this is not convincing. When a densitometic analysis is carried out, does this analysis confirm increase at of phosphorylated AKT in CVA2 treated animals at day 3 and 7?

  1. Line 267 Authors state, that “heart injury induced by CVA2 has not been reported in mouse model”. This is wrong, as their recent published paper show that CVA2 induces heart injury. Please change.
  2. Line 285 Authors state, “It still had a significant effect on cardiac function” as mentioned above, this is not correct as cardiac function by hemodynamic measurement was not carried out.

Author Response

Q1: There was a previous paper of the group published in Frontiers in Microbiology where authors used the same CVA2, variant as in the present manuscript. In this previous paper I tried to open the link for the virus HN202009, but found “ AL_NL060I15.R AL_NL Arabidopsis lyrata genomic 3', genomic survey sequence”. Moreover, I did not found an entry for the accession number MT992622 for HN202009. Please insert into the manuscript a link where the accession number can be found.

We also found this problem. We will notify the editor to make corrections later. HN202009 is the isolate ID for this viral sequence. The data have become available from Genbank now (https://www.ncbi.nlm.nih.gov/nuccore/MT992622).

Q2: Figure 1C I am not sure, whether the cells which show brown staining are cardiomyocytes. At least the left cell (left arrow) could also be a macrophage. For the right arrow I am not sure what it shows. Coxsackieviruses can also infect immune cells and other cells, e.g. endothelial cells. To confirm that the VP1 signal is indeed located in cardiomyocytes a co-staining with cardiomyocyte specific antibody should be done. Moreover, it would be helpful to examine whether cardiomyocytes are targets of the virus by infection of the cells in vitro with the CVA2 strain.

We have simultaneously conducted immunofluorescence staining of viral VP1 proteins and cardiomyocytes (cardiac troponin I), and found VP1 signal was indeed located in cardiomyocytes. The cardiomyocytes are one of the target cells, which may directly lead to heart injury.

Q3: Line 108 Please specify the instrument using for real-time PCR

We have revised it.

Q4: Line 192 to 196, 210. I believe the authors mean CVA2 and not EV71?

We have revised it.

Q5: Figure 2E the WB should be quantified and the data shown as diagram.

We have revised it.

Q6: Line 213 replace “cardiomyocytes” by heart, as the whole heart was investigated.

We have revised it.

Q7: Line 196 the authors state that the data indicate cardiac dysfunction. This term is typically used to describe impaired contractility of the heart, which was not determined here. Thus, another term should be use e.g. dysregulation of cardiac proteins. Please also replace “cardiac function” in line 189 by another therm.

We have revised it.

Q8: The histological data described in the text (line 172-176) should be better explained in the figure 1E. Please explain blue staining (day 7 CVA2). Is it collagen, does it show fibrosis? The authors mention occurrence of damage of cardiomyocytes. Insert arrows for such events.

We have revised it.

Q9: Although pathological alteration as shown in Fig. 1E seems to be not as prominent as at day 7 the data from day 3 should be described in the text.

We have revised it.

Q10: Line 220, replace Figure by Figure 3

We have revised it.

Q11: Figure 3D the WB seems not confirm some of the data obtained with qRT-PCR and IF, such as I do not see increase of MMP8 in CVA2-infected mice in WB, it seems rather to be decreased. TIMP4 expression levels are similar in untreated and CVA2-infected mice in WB, whereas it is distinctly higher expressed in CVA2-infected mice as shown by IF and qRT-PCR. As also the loading control is highly variable, a quantification of WB should be done and the data shown in a diagram.

We have repeated the experiments or reduced the background, and provided more clear bands for TIMP4, MMP8, 9 and 3 and CTGF and shown the data as diagram.

Q12: Line 229-234 Based on measurement of certain biomarkers of endothelial dysfunction the authors state that CVA2 infected mice have endothelial dysfunction. There are no clinical data determining endothelial dysfunctions. Accordingly, their statement should be adapted.

There have many clinical data reporting endothelial dysfunctions or endothelial activation in patients with hand-foot-mouth disease. Particularly, the importance of some important biomarkers (Vwf, VEGF, VCAM-1 and so on) in HFMD patients have been reported. The occurrence of hand-foot-and-mouth disease complicated with encephalitis children has correlation with the expressions of VCAM-1 and VEGF in serum and cerebrospinal fluid and the levels has correlation with the state of illness (Chen Jiao,Lei Zhixian,Chen Beibei.The detective significance of VCAM- and VEGF in hand-foot-mouth disease complicated with encephalitis children [J].China Maternal and Child Health Care,2015,30(19):3186-3188.). We are the first to report endothelial cell injury of the heart with HFMD. The mechanism of endothelial cell injury and its relationship with CVA2-induced cardiac injury need to be further studied.

Q13: Line 249. Authors state that inflammatory -related pathways were induced. From the WB for AKT this is not convincing. When a densitometic analysis is carried out, does this analysis confirm increase at of phosphorylated AKT in CVA2 treated animals at day 3 and 7?

When a densitometric analysis is carried out, the phosphorylation (activation) of Akt in CVA2 treated animals was enhanced at 3 dpi but almost no change at 7 dpi. This means that phosphorylation Akt is probably one of the sources of cytokines during disease progression rather than the late phase.

Q14: Line 267 Authors state, that “heart injury induced by CVA2 has not been reported in mouse model”. This is wrong, as their recent published paper show that CVA2 induces heart injury. Please change.

We have revised it.

Q15: Line 285 Authors state, “It still had a significant effect on cardiac function” as mentioned above, this is not correct as cardiac function by hemodynamic measurement was not carried out.

We have revised it.

Round 2

Reviewer 2 Report

CVA2 induced heart injury cannot be presented in this article without as control CVB3 Nancy our CVB3/28 infection with several temporal points at the same time post inoculation in mice. Major limits about the direct pathogenic role of cvA2 are displayed in the presented data ; no or very slight CVA2 heart myocardial lesions  (see the dallas criteria AHA references) (figure 1 ; no necrosis ; slight inflammation) at 3pi and at 7 pi low levels of lesions. Moreover the levels of viral production , PFU per mg of heart tissues are lower than previously published with CVB3 (Andreoletti et al; Chapman et al; Tracy et al; ; see related Pubmed references) . Concerning the heart vessels and endothelial cells the presented work is not logically presented and ii is only the beginning of a further new report which have been to be developed before a further submission. Ethical aspects of the animal experimentation remains to be clarified and are lacking yet in this new corrected version. CVA2 induced heart injury cannot be presented in this article without as control CVB3 Nancy our CVB3/28 infection with several temporal points at the same time post inoculation in mice. Major limits about the direct pathogenic role of cvA2 are displayed in the presented data ; no or very slight CVA2 heart myocardial lesions  (see the dallas criteria AHA references) (figure 1 ; no necrosis ; slight inflammation) at 3pi and at 7 pi low levels of lesions. Moreover the levels of viral production , PFU per mg of heart tissues are lower than previously published with CVB3 (Andreoletti et al; Chapman et al; Tracy et al; .....; see related Pubmed references) . Concerning the heart vessels and endothelial cells experimental findings , the order of figures is not logically presented. For me it s only the beginning of a further new report which have been to be developed before a further submission. Ethical aspects of the animal experimentation remains to be clarified and are lacking yet in this new corrected version

Author Response

Q1: CVA2 induced heart injury cannot be presented in this article without as control CVB3 Nancy our CVB3/28 infection with several temporal points at the same time post inoculation in mice. Major limits about the direct pathogenic role of cvA2 are displayed in the presented data; no or very slight CVA2 heart myocardial lesions (see the dallas criteria AHA references) (figure 1; no necrosis; slight inflammation) at 3pi and at 7 pi low levels of lesions.

A1: We know that some patients with hand, foot, and mouth disease (HFMD) died with acute heart disease (always reported as viral myocarditis). In this study, we used a neonatal mouse model to study the possible mechanisms of CVA2-induced heart injury, which is of great significance for understanding the pathogenesis of HFMD, and providing therapeutic targets for the treatment of CVA2-associated heart injury.

We agree with the reviewer that we need to lower the tones of writing. Dallas criteria AHA references are about diagnosis of myocarditis. Thus, we have changed “myocarditis” to “heart injury”. It is known that CVB3 is very dangerous and can cause fulminant myocarditis. It is true that CVA2 strain does not have a strong cardiac tropism like CVB3, and our study only aims to demonstrate that CVA2 infection can lead to heart injury, rather than to compare the differences between CVB3 and CVA2-induced heart lesions. In addition, CVB3-induced myocarditis animal model uses adult mice, but our animal model uses neonatal mice. In human infections, CVB3-induced disease always comes from adult male individuals, while CVA2-induced disease mainly comes from infants and young children. Therefore, we think that while it is important to compare these two strains as the reviewer suggested, it is not necessary to use the CVB3 infection group as a control for our purpose. CVB3 infection we mentioned in this paper only provides a reference for us to elucidate the possible mechanisms of heart injury caused by CVA2 infection.

Normally, an injury to the heart is often characterized as fibrosis, inflammation, with the deposition of additional extracellular matrix proteins and the increase of inflammatory responses (PMID: 32252591; PMID: 32454355; PMID: 25677476). Based on the results of pathological examination, heart injury caused by CVA2 is not as typical and severe as that caused by CVB3 infection, but pathological changes did appear. The alterations of myocardial enzymes support the pathological changes induced by CVA2. We also found the deposition of additional extracellular matrix proteins (fibrosis), leukocyte infiltration and excessive inflammatory responses (inflammation), which further demonstrate heart injury after CVA2 infection. Unlike SARS-CoV-2-induced heart injury and PM2.5 (air pollutants)-induced heart injury, we found viral replication in heart tissues, indicating direct injury of cardiomyocytes.

Q2: Moreover, the levels of viral production, PFU per mg of heart tissues are lower than previously published with CVB3 (Andreoletti et al; Chapman et al; Tracy et al; see related Pubmed references).

A2: We agree with the reviewer’s point. Both of CVA2 and CVB3 belong to enteroviruses. It is well known that CVB3 is a typical cardiac tropism virus, which is more likely to proliferate in the heart resulting in a high viral load. According to our experimental results, CVA2 can infect cardiomyocytes, but the proliferation in the heart is limited compared with CVB3. These invading CVA2 viruses finally lead to heart injury.

Q3: Concerning the heart vessels and endothelial cells the presented work is not logically presented and it is only the beginning of a further new report which have been to be developed before a further submission.

A3: We have deleted the data of endothelial dysfunction.

Q4: Ethical aspects of the animal experimentation remains to be clarified and are lacking yet in this new corrected version.

A4: line 482-487 In the section of “Institutional Review Board Statement”, we have provided ethical statements concerning mice model (protocol code: ZZUIRB2020-29, date of approval: April 2019).

Reviewer 3 Report

The manuscript has improved significantly and my questions have been answered convincingly.

However I have still two questions.

  1. line 334 the authors wrote: "We detected CVA2 nucleus and antigen in heart tissues...."  I do not understand this senctence. What they mean with CVA2 nucleus? The virus does not replicate in the nucleus. Please correct/explain.
  2.  line 419: the authors state" We observed markedly monocytes infiltration in the heart tissue of infected mice". There are no FACS data showing that monocytes infiltate the heart after CVA2 infection. Also the histological examination (Fig.1E) does not confirm "markedly" infiltration. Thus for me it is not clear, how the authors come to this conclusion. Please explain.

Author Response

Q1: line 334 the authors wrote: "We detected CVA2 nucleus and antigen in heart tissues...."  I do not understand this senctence. What they mean with CVA2 nucleus? The virus does not replicate in the nucleus. Please correct/explain.

We have changed “nucleus” to “VP1 mRNA”.

Q2: line 419: the authors state" We observed markedly monocytes infiltration in the heart tissue of infected mice". There are no FACS data showing that monocytes infiltate the heart after CVA2 infection.

As we all know, FACS can be used to count cells based on the specific markers on the cell surface. However, in our current research, we used immunofluorescence staining to label monocytes (CD11b+) in heart slices. This method can not only locate monocytes but also can be applied to quantify the number of monocytes (PMID: 31534976; PMID: 30661445). In this study, we quantified the number of monocytes in heart slices by using Image J software. Our results showed that the number of monocytes per unit area in the heart slices of infected mice was significantly increased compared to controls. Therefore, we concluded that monocytes infiltration was observed in the heart slices of infected mice.

Q3: Also, the histological examination (Fig.1E) does not confirm "markedly" infiltration. Thus for me, it is not clear, how the authors come to this conclusion. Please explain.

We agree with the reviewer that we need to lower the tones of writing and have removed “markedly” from our text.